# Iridium-Catalyzed Highly Selective 1,4-Reduction of α,β-Unsaturated Carbonyl Compounds

**DOI:** 10.3390/molecules29245912

**Published:** 2024-12-14

**Authors:** Youwei Chen, Jide Li, Jiaxi Xu, Zhanhui Yang

**Affiliations:** Department of Organic Chemistry, College of Chemistry, Beijing University of Chemical Technology, Beijing 100029, China; 2023200930@buct.edu.cn (Y.C.); 18724730682@163.com (J.L.); jxxu@mail.buct.edu.cn (J.X.)

**Keywords:** iridium catalysis, transfer hydrogenation, alkenes, formic acid, water solvent

## Abstract

In this study, an iridium-catalyzed selective 1,4-reduction of α,β-unsaturated carbonyl compounds is realized, with water as a solvent and formic acid as a hydride donor. The new efficient iridium catalyst features a 2-(4,5-dihydroimidazol-2-yl)quinoline ligand. The chemoselectivity and catalyst efficiency are highly dependent on the electronic and steric properties of the substrates. For α,β-unsaturated amides, acids, and esters, only the electron-deficient C=C bonds are reduced (1,4-reduction), and the other functional groups are left intact. The *S/C* ratio and initial TOF reach 7000 and 18,480 h^−1^, respectively. A gram-scale 1,4-reduction is also performed. Deuterium labeling shows that the β-hydrogens of the products originate from the formyl hydrogen in the formic acid. The application of the 1,4-reduction for the modification the structures of some medications is demonstrated.

## 1. Introduction

The 1,4-reduction of α,β-unsaturated carbonyl compounds represents one of the most efficient methods to prepare carbonyl derivatives, and finds wide applications in fine chemical and pharmaceutical production [1], as well as in natural product synthesis [2,3]. As a consequence, much attention has been paid to this field. Well-developed stochiometric reducing reagents include metal (boro)hydrides [4,5,6,7,8,9,10,11], dissolving metals [12,13], biological reducing agents [14], and hydrogen-donating heterocycles [15,16,17], etc. However, transition-metal-catalyzed hydrogenation (Figure 1a) [18,19,20,21,22] and transfer hydrogenation (TH) (Figure 1b) [23,24,25,26,27,28], especially the latter, still constitute an intriguing direction. Several catalyst–ligand combinations have been reported, with different hydride donors (triethyl ammonium formate, isopropanol, silane, etc.), to achieve the 1,4-reduction of α,β-unsaturated carbonyl compounds in organic solvents under basic or neutral conditions (Figure 1b). In spite of these advances and the recent state of the art [18,22,27], a highly efficient TH 1,4-reduction under acidic conditions in water is still in demand [29,30] for the following reasons: (1) metal (boro)hydride reduction (for example NaBH_4_, LiAlH_4_) is often conducted under basic conditions, making it incompatible with compounds with base-sensitive functionalities such as carboxylic acid, and (2) compared with neutral conditions, acidic conditions can further activate the substrate and thus facilitate the reductions.

We previously reported a novel series of iridium catalysts with 2-(4,5-dihydroimidazol-2-yl)pyridine derivatives as ligands [31,32,33,34,35,36,37,38,39,40,41,42,43]. These catalysts were harnessed to realize the highly efficient and selective aerobic oxidation of aldehydes [31], the transfer hydrogenation of aldehydes [32], ketones [33], nitroalkenes [36,37], and imines [38], the deoxygenation of alkanols [34,39,40], the reduction–elimination of steroidal 4-en-3-ones [35], and other reactions [41,42,43], with water as the (co)solvent [44,45,46]. In the above reductions, iridium hydrides, generated from formic acids, were identified as the key reductive species to transfer hydrides to electrophilic carbonyls and carbocations. Herein, we additionally demonstrated a similar hydride transfer to electron-deficient C=C double bonds from the iridium hydrides, thus providing an efficient and green method to prepare carbonyl-functionalized alkanes in water with formic acid as a traceless hydride donor (Figure 1c). The extension of the viable scope from C=O bonds to C=C bonds represents a significant step for our catalysts.

## 2. Results and Discussion

We used the reaction of 3-acryloyloxazolidin-2-one (**1a**) as a model to optimize the conditions (Table 1). An initial screening of the catalysts (entries 1–11) at an *S/C* ratio of 700 revealed that **C1**, **C2**, **C5**, **C10**, and **C11** showed competent abilities to completely convert **1a** to 3-propionyloxazolidin-2-one (**2a**) (>99% conversion). Distinguishing these catalysts at a higher *S/C* ratio of 1400 (entries 12–16) showed **C1** as the most efficient catalyst. Decreasing the catalyst loading of **C1** delivered lower yields (entries 17 and 18). However, the results were still striking. At 3500 and even 7000 *S/C* ratios, **C1** still showed a good activity, and the conversion rates were 87% and 55%, respectively. The optimization of the amounts of formic acid (entries 19–22) indicated that the reduction of **1a** with **C1** was very feasible, and three equivalents of formic acid were enough to render a complete conversion (entry 22). It must be noted that the reduction of **1a** showed a very good chemoselectivity, and **2a** was an exclusive product in all cases.

For our previously synthesized catalysts (**C2**–**C8**), their electronic properties were tuned by the substituents on the pyridyl rings. However, for the catalyst **C1**, a quinolinyl ring was directly introduced to modulate the electronic density of the metal center, and it showed a very high efficiency in the transfer hydrogenation of **1a** with formic acid. We also measured the TOF value of **C1** against time for the reduction of **1a**, and it was as high as 18,480 h^−1^ (Table 2).

With the optimized catalyst, we tried to examine its activity in the reduction of structurally different α,β-unsaturated carbonyl compounds (Figure 1). The results indicated that the catalyst’s efficiency was closely associated with the electronic and steric features of substrates. To obtain a satisfactory yield, several different sets of conditions (Conditions A–E) were tried. For Conditions D and E, the formic acid was added portion-wise because our previous work revealed that the catalyst’s efficiency would be lowered by the strong acidity rendered by the large amount of formic acid [32,33]. By adopting portion-wise addition, the second portion was added following the consumption of the first portion, thus keeping the reaction medium less acidic.

Although **1a** was readily reduced at the high *S/C* ratio (Table 1, entries 17–22), the substrates listed in Figure 1 required an increased catalyst usage and a higher amount of formic acid because the electronic and steric features of substrates dramatically varied. We first exploited the reduction of acrylamide derivatives. Improving catalyst dosage (*S/C* = 50) and formic acid amount (64 equiv.), together with prolonging reaction time (3 h), gave satisfactory results. Complete (**1b**, **1c**, **1d**, **1g**, and **1j**) or considerably improved conversions (**1e**, **1f** and **1i**) were observed by ^1^H NMR determination. In addition, 3-phenylacrylic acid (**1k**) and 2-phenylacrylic acid (**1l**) were reduced with 81% (*S/C* = 25) and 100% (*S/C* = 100) conversions, respectively. Excitingly, the carboxylic group found in **2k** and **2l** was well tolerated under the current acidic conditions. Further asymmetric reduction of analogs of **1l** with chiral catalysts of this type is in progress, as these products are important pharmaceutical intermediates (for example, Naproxen). Ethyl cinnamate (**1m**) was reduced with a 47% conversion under Conditions D. The installation of two ester groups at the same carbon largely increased its reactivity toward the 1,4-reduction (**2n**). The reduction of diethyl maleate (**1o**) at a 50 *S/C* ratio delivered a 71% conversion. Substrates containing other electron-withdrawing groups such as cinnamonitrile (**1p**) and vinylsulfones (**1q**) showed exceedingly low activities, presumably because the cyano and sulfonyl groups are difficult to protonate to activate the corresponding C=C bonds. As shown by the ^1^H NMR of the crude reaction mixtures, in all of the reactions with incomplete conversions (**2i**, **2m**), no side-products were observed, and only the unreacted substrate and product were observed.

We also found that the iridium-catalyzed transfer hydrogenation of enones (**1r**, **1s**) and enals (**1t** and **1u**) gave over-reduced products (Figure 2). The reduction of chalcone (**1r**) delivered ketone **2r**, alkanol **2ra**, and allylic alcohol **2rb** in 60%, 17%, and 23% yields, respectively (eqn. 1). The reduction of **1s** gave ketone **2s** and alkanol **2sa** in respective 18% and 75% yields (eqn. 2), while the reduction of methacrylaldehyde (**1t**) and cinnamic aldehyde (**1u**) exclusively produced alkanols **2ta** and **2ua** (eqn. 3 and eqn. 4). The formation of **2ra**, **2sa**, **2ta,** and **2ua** was attributed to the over-reduction of **2r**, **2s**, **2t**, and **2s**, respectively. In these cases, the initial chemoselective ratios (C=C reduction vs. C=O reduction) were 77:23 for **1r**, 100:0 for **1s**, **1t**, and **1u**, indicating that the reduction of C=C bonds took priority over C=O bonds, although our previous cinnamic aldehyde reduction showed the opposite selectivity [32]. Such a 1,4-chemoselectivity was also implied in our previous report [33].

Since α,β-unsaturated carbonyl structures are omnipresent structural subunits in many medicinal and natural molecules, we sought to use our protocol for the modifications of these structures (Figure 3). Surprisingly, **C1** showed inferior activity in these cases. However, the use of **C4** gave satisfactory results [35]. Santonin (**3a**) is an anthelminthic widely used in the past to expel parasitic worms from the body. The exposure of **3a** to the reduction conditions resulted in transfer hydrogenation at only C6=C7 double bonds, producing **4a** with a 65% yield. The electron-deficient and sterically large C9=C10 double bond was left intact. Metandienone (**3b**) and Prednisone (**3c**) are both steroid medications. The former was once used for the treatment of hypogonadism, and is now used as a performance-enhancing drug, while the latter is mostly used to suppress the immune system and decrease inflammation. They first underwent C1=C2 bond reduction, followed by the reduction of C=O bonds and dehydration to produce the steroidal dienes **4b** and **4c** in 35% and 29% yields, respectively [35]. Similarly, the electron-deficient 6-methylene of **3d** was first transfer hydrogenated, and subsequent transformations delivered diene **4d** in a 77% yield [35]. It should be mentioned that molecules **3a–d** each have two sterically different electron-deficient C=C bonds, but only the sterically lowest one was reduced.

The iridium-catalyzed 1,4-reduction is easy to scale up. For example, 0.86 g of **1a** was readily reduced at a 1400 *S/C* ratio in an open flask, and product **2a** was isolated with a 91% yield by extraction with ethyl acetate, drying, and concentration under reduced pressure (Figure 4). GC-MS analysis shows the good purity of the product (see Appendix A).

To gain insight into the reaction mechanism, we performed deuterium-labeling experiments (Figure 5). When using DCO_2_D instead of HCO_2_H, we observed a 6% deuterium incorporation ([D]) at the α-position and >99% incorporation at the β-position (Figure 5a). This suggests that the β-hydrogen in the products originates from formic acid. When using HCO_2_D instead of HCO_2_H, we observed 6% α-[D] and 3% β-[D] (Figure 5b). This suggests that the hydrogen in β-position comes from the formyl group of formic acid. When using D_2_O instead of H_2_O, we observed >99% α-[D] and 5% β-[D] (Figure 5c). The former is attributed to the well-known rapid H–D exchange at the active α-positions of carbonyls via enol–ketone tautomerization, while the latter indicates a less favored H–D exchange between [Ir]-H and D_2_O.

Based on our previous work [33,34,35,36,37,38,39,40,41,42,43] and the deuterium-labeling experiments, a plausible mechanism for the 1,4-reduction of α,β-unsaturated carbonyl compounds was proposed (Figure 6). Iridium formate (**A**) (step i) is first generated, the β-elimination of which gives rise to iridium hydride (**B**) as a catalyst in a resting state and a transient reducing reagent (step ii). Similar iridium hydride was demonstrated by ^1^H NMR in our previous work [33]. On the other hand, under acidic conditions, α,β-unsaturated carbonyl compounds (for example, **1a**) are activated by protonation of the carbonyl group (**5a**). Subsequently, the 1,4-addition of iridium hyride (**B**) to activated **5a**, in an outer-sphere fashion, in the presence of formic acid produces enol (**6a**) and regenerates iridium formate **A** (step iii). The final product (**2a**) is generated via enol-to-ketone tautomerization (step v). In the presence of HCO_2_D, DCO_2_D, or D_2_O, the H–D exchange (step iv and step v) will impact the deuterium incorporations ([D]) in the products; namely, step iv determines β-[D], and step v affects α-[D].

## 3. Materials and Methods

### 3.1. Materials and Instruments

Unless otherwise noted, all materials were purchased from commercial suppliers. Dichloromethane (DCM) and chlorobenzene were refluxed over CaH_2_; tetrahydrofuran (THF) and toluene were refluxed over metal sodium. The solvents were freshly distilled prior to use. Column chromatography was performed on silica gel (normal phase, 200–300 mesh) from Anhui Liangchen Silicon Material Co., Ltd., with petroleum ether (PE, bp. 60–90 °C) and ethyl acetate (EA) and petroleum ether (PE) as eluents. Reactions were monitored by thin-layer chromatography (TLC) on GF_254_ silica gel plates (0.2 mm) from Anhui Liangchen Silicon Material Co., Ltd. (Lu’An, Anhui Province, China) The plates were visualized by UV light, and other TLC stains (1% potassium permanganate in water; 10 g of iodine absorbed on 30 g of silica gel). ^1^H and ^13^C NMR spectra were recorded on a Bruker 400 MHz spectrometer, usually with CDCl_3_ as an internal standard, and the chemical shifts (*δ*) were reported in parts per million (ppm). Multiplicities are indicated as s (singlet), d (doublet), t (triplet), q (quartet), dd (double doublet), m (multiplet), and b (broad). Coupling constants (*J*) are reported in Hertz (Hz). Melting points were obtained on a SGW X-4A melting point apparatus and were uncorrected.

All the materials and products are known compounds, and their ^1^H and/or ^13^C NMR are identical to those reported.

The catalysts **C2**–**C8** were prepared according to our previous publications [32,33], and **C9**–**C11** according to Li and co-workers’ publications [47,48]. The catalyst solution was prepared according to our previous publication [32].

### 3.2. Preparation of the Catalyst C1

**C1** was prepared according to our previous procedure [36]. Ethylenediamine (0.16 mL, 2.3 mmol) was added dropwise to the solution of 2-quinolinecarboxaldehyde (2.1 mmol) in CH_2_Cl_2_ (20 mL). The mixture was stirred for 1 h. *N*-Bromosuccinimide (410 mg, 2.3 mmol) was added in three portions at 0 °C and the mixture was stirred overnight. The reaction mixture was washed with 5% NaOH solution (20 mL) and saturated Na_2_S_2_O_3_ solution (20 mL), and dried with sodium sulfate. The solvent was removed in vacuo. The resultant crude product 2-(4,5-dihydro-1*H*-imidazol-2-yl)quinoline was dissolved in 20 mL of dry CH_2_Cl_2_, and then [Cp*IrCl_2_]_2_ (720 mg, 0.9 mmol) was added. The solution was stirred overnight. The solvent was removed in vacuo, and the resultant red solid was dissolved in a minimal amount of CH_2_Cl_2_. Then, a large amount of EtOAc was added to precipitate a red solid as the desired product, which was filtered and further dried under vacuum. Yield: 595 mg, 62%. ^1^H NMR (400 MHz, D_2_O) δ 8.58 (d, *J* = 8.4 Hz, 1H), 8.34 (d, *J* = 9.2 Hz, 1H), 8.09–8.02 (m, 2H), 7.88 (d, *J* = 8.4 Hz, 1H), 7.84 (dd, *J* = 8.0, 7.0 Hz, 1H), 4.51–4.36 (m, 1H), 4.15 (t, *J* = 10.4 Hz, 2H), 4.06–3.92 (m, 1H), 1.60 (s, 15H); ^13^C NMR (101 MHz, D_2_O) δ 169.9, 146.9, 145.4, 141.7, 133.0, 130.34, 130.30, 129.5, 129.2, 119.6, 89.2, 52.3, 45.7, 8.5.



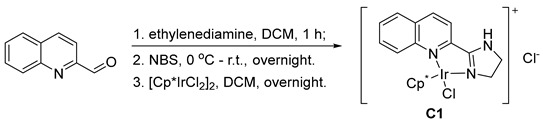



### 3.3. Optimization of Reaction Conditions

#### 3.3.1. Optimization of Catalysts in Entries 1–11, Table 1

3-acryloyloxazolidin-2-one (**1a**, 20 mg, 0.14 mmol) and 0.4 mL of catalyst solution (0.0005 mol/L, **C1**–**C11**) in deionized water were added to a 10 mL reaction tube. The mixture was stirred for 5 min at 80 °C, and then formic acid (122 μL, 23 equiv.) was added. After stirring for 1 h in open air, the mixture was diluted with water (4 mL) and extracted with ethyl acetate (3 mL × 3). The organic solvent was evaporated under reduced pressure and the crude residue was submitted to ^1^H NMR conversion determination.

#### 3.3.2. Optimization of Catalyst Type in Entries 12–16, Table 1

3-acryloyloxazolidin-2-one (20 mg, 0.14 mmol) and 0.4 mL of catalyst solution (0.00025 mol/L, **C1**, **C2**, **C5**, **C10** or **C11**) in deionized water were added to a 10 mL reaction tube. The mixture was stirred for 5 min at 80 °C, and then formic acid (122 μL, 23 equiv.) was added. After stirring for 1 h in open air, a similar workup to that outlined above was carried out.

#### 3.3.3. Optimization of Catalyst Loading in Entries 17–18, Table 1

3-acryloyloxazolidin-2-one (20 mg, 0.14 mmol) and 0.4 mL of catalyst **C1** solution (0.0001 mol/L or 0.00005 mol/L) in deionized water were added to a 10 mL reaction tube. The mixture was stirred for 5 min at 80 °C, and then formic acid (122 μL, 23 equiv.) was added. After stirring for 1 h in open air, a similar workup to that outlined above was carried out.

#### 3.3.4. Optimization of Equivalents of Formic Acid in Entries 19–22, Table 1

3-acryloyloxazolidin-2-one (20 mg, 0.14 mmol), 0.4 mL of catalyst **C1** solution (0.00025 mol/L) in deionized water was added to a 10 mL reaction tube. The mixture was stirred for 5 min at 80 °C, and then formic acid (15 μL, 3 equiv.; 31 μL, 6 equiv; 61 μL, 11 equiv. or 92 μL, 17 equiv.) was added. After stirring for 1 h in open air, a similar workup to that outlined above was carried out.

### 3.4. Procedure for Measuring TOF Values

3-acryloyloxazolidin-2-one (20 mg, 0.14 mmol) and 0.4 mL of catalyst **C1** solution (0.00005 mol/L) in deionized water were added to a 10 mL reaction tube. The mixture was stirred for 5 min at 80 °C, and then formic acid (122 μL, 23 equiv.) was added. A similar workup to that outlined above was carried out for 5 min, 10 min, 30 min, 60 min, and 120 min in open air.

### 3.5. General Procedure for Synthesis of Acrylamide [49]

Potassium carbonate (662 mg, 4.8 mmol) was added to a solution of amine (4.0 mmol) in dry acetonitrile (40 mL) at 0 °C. After stirring for 30 min, acryloyl chloride (0.4 mL, 4.8 mmol) was added dropwise to the mixture. The reaction mixture was stirred at room temperature. The reaction mixture was filtrated through Celite. The filtrate was evaporated in vacuo and the crude product was purified by column chromatography on silica gel with petroleum ether (PE, 60–90 °C fraction) and ethyl acetate (EA) as the eluent.



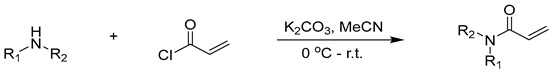



#### 3.5.1. N,N-Diethylacrylamide (**1c**)

Reaction time: 12 h, R*_f_* = 0.3 (PE/EA = 1:1, *v*/*v*), colorless liquid, yield: 69%; ^1^H NMR (400 MHz, CDCl_3_) δ 6.55 (dd, *J* = 16.7, 10.4 Hz, 1H), 6.34 (dd, *J* = 16.8, 2.1 Hz, 1H), 5.67 (dd, *J* = 10.4, 2.1 Hz, 1H), 3.45 (q, *J* = 7.2 Hz, 2H), 3.39 (q, *J* = 7.2 Hz, 2H), 1.20 (t, *J* = 7.2 Hz, 3H), 1.16 (t, *J* = 7.2 Hz, 3H); ^13^C NMR (101 MHz, CDCl_3_) δ 165.6, 127.9, 127.5, 42.2, 40.8, 14.8, 13.1.

#### 3.5.2. N,N-Diethylacrylamide (**1d**)

Reaction time: 12 h, R*_f_* = 0.1 (PE/EA = 1:1, *v*/*v*), colorless liquid, yield: 81%; ^1^H NMR (400 MHz, CDCl_3_) δ 6.46 (dd, *J* = 16.8, 10.0 Hz, 1H), 6.35 (dd, *J* = 16.8, 2.4 Hz, 1H), 5.66 (dd, *J* = 10.0, 2.4 Hz, 1H), 3.54 (t, *J* = 6.8 Hz, 4H), 1.98 (quintet, *J* = 6.7 Hz, 2H), 1.88 (quintet, *J* = 6.8 Hz, 2H); ^13^C NMR (101 MHz, CDCl_3_) δ 164.2, 128.7, 126.9, 46.4, 45.6, 25.9, 24.1.

#### 3.5.3. N,N-Dibenzylacrylamide (**1e**)

Reaction time: 6 h, R*_f_* = 0.1 (PE/EA = 10:1, *v*/*v*), colorless liquid, yield: 62%; ^1^H NMR (400 MHz, CDCl_3_) δ 7.39–7.23 (m, 8H), 7.16 (d, *J* = 7.4 Hz, 2H), 6.61 (dd, *J* = 16.7, 10.2 Hz, 1H), 6.49 (dd, *J* = 16.6, 2.2 Hz, 1H), 5.72 (dd, *J* = 10.2, 2.2 Hz, 1H), 4.65 (s, 2H), 4.51 (s, 2H); ^13^C NMR (101 MHz, CDCl_3_) δ 166.9, 137.1, 136.4, 129.0, 128.9, 128.6, 128.3, 127.62, 127.55, 127.4, 126.4, 49.8, 48.5.

#### 3.5.4. N-Methyl-N-Phenylacrylamide (**1f**)

Reaction time: 4 h, R*_f_* = 0.2 (PE/EA = 5:1, *v*/*v*), light yellow crystals, yield: 95%, ^1^H NMR (400 MHz, CDCl_3_) δ 7.45–7.38 (m, 2H), 7.37–7.29 (m, 1H), 7.21–7.16 (m, 2H), 6.37 (dd, *J* = 16.8, 2.0 Hz, 1H), 6.08 (dd, *J* = 16.8, 10.3 Hz, 1H), 5.51 (dd, *J* = 10.3, 2.0 Hz, 1H), 3.36 (s, 3H); ^13^C NMR (101 MHz, CDCl_3_) δ 165.6, 143.3, 129.4, 128.4, 127.5, 127.2, 127.1, 37.3.

#### 3.5.5. N,N-Diphenylacrylamide (**1g**)

Reaction time: 8 h, R*_f_* = 0.2 (PE/EA = 10:1, *v*/*v*), colorless crystals, yield: 41%; ^1^H NMR (400 MHz, CDCl_3_) δ 7.41–7.32 (m, 4H), 7.30–7.19 (m, 6H), 6.47 (dd, *J* = 16.8, 2.0 Hz, 1H), 6.19 (dd, *J* = 16.8, 10.3 Hz, 1H), 5.62 (dd, *J* = 10.3, 2.0 Hz, 1H); ^13^C NMR (101 MHz, CDCl_3_) δ 165.7, 142.4, 129.5, 129.2, 128.3, 126.9 (m).

### 3.6. Synthesis of (Vinylsulfonyl)benzene (***1q***) [50]

NaO^t^Bu (154 mg, 1.6 mmol) was added to a solution of thiirane (53 μL, 1.0 mmol) and diphenyliodonium chloride (316 mg, 1.0 mmol) in dry THF (12 mL). The mixture was stirred at room temperature for 4 h. After the solvent was removed, 20 mL of water was added and the mixture was extracted with CH_2_Cl_2_ (10 mL × 3). The combined organic phase was concentrated under reduced pressure. The resulting yellow liquid was diluted with CH_2_Cl_2_ (15 mL), and was then added dropwise to a performic acid solution, prepared by mixing and stirring 30% H_2_O_2_ (0.7 mL, 6.0 mmol) and 88% HCO_2_H (6.0 mL) at room temperature for 0.5 h, at 0 °C. The mixture was stirred at room temperature for 2 h, and then was washed with water (10 mL × 3). The organic phase was dried over Na_2_SO_4_ and concentrated in vacuo. The crude product was purified by column chromatography on silica gel to produce a white solid (66 mg, 39%), *R_f_* = 0.2 (PE/EA, 10:1);^1^H NMR (400 MHz, CDCl3): δ 7.82 (d, *J* = 7.8 Hz, 2 H), 7.57 (t, *J* = 7.1 Hz, 1 H), 7.48 (t, *J* = 7.8 Hz, 2 H), 6.59 (dd, *J* = 9.6, 16.5 Hz, 1 H), 6.39 (d, *J* = 16.6 Hz, 1 H), 5.97 (d, *J* = 9.7 Hz, 1 H); ^13^C NMR (101 MHz, CDCl_3_) δ 139.5, 138.4, 133.6, 129.3, 127.9, 127.7.



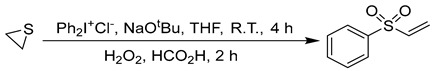



### 3.7. General Procedure

Reaction Conditions A: α,β-unsaturated carbonyl compounds (0.2 mmol) and 0.4 mL of catalyst **C1** solution (0.00025 mol/L) in deionized water were added to a 10 mL reaction tube. The mixture was stirred for 5 min at 80 °C, and then formic acid (15 μL, 2 equiv.) was added. After stirring for 1 h in open air, the mixture was diluted with water (4 mL) and extracted with ethyl acetate (3 mL × 3). The solvent was removed under reduced pressure and the crude residue underwent ^1^H NMR yield determination.

Reaction Conditions B: α,β-unsaturated carbonyl compounds (0.2 mmol) and 0.4 mL of catalyst **C1** solution (0.0025 mol/L) in deionized water were added to a 10 mL reaction tube. The mixture was stirred for 5 min at 80 °C, and then formic acid (61 μL, 16 equiv.) was added. After stirring for 1 h in open air, the mixture was diluted with water (4 mL) and extracted with ethyl acetate (3 mL × 3). The solvent was removed under reduced pressure and the crude residue underwent ^1^H NMR yield determination.

Reaction Conditions C: α,β-unsaturated carbonyl compounds (0.1 mmol) and 0.4 mL of catalyst **C1** solution (0.0025 mol/L) in deionized water were added to a 10 mL reaction tube. The mixture was stirred for 5 min at 80 °C, and then formic acid (61 μL, 16 equiv.) was added. After stirring for 1 h in open air, the mixture was diluted with water (4 mL) and extracted with ethyl acetate (3 mL × 3). The solvent was removed under reduced pressure and the crude residue underwent ^1^H NMR yield determination.

Reaction Conditions D: α,β-unsaturated carbonyl compounds (0.1 mmol), 0.4 mL of catalyst **C1** solution (0.005 mol/L) in deionized water were added to a 10 mL reaction tube. The mixture was stirred for 5 min at 80 °C, and then formic acid (122 μL, 32 equiv.) was added. After stirring for 1 h, a second portion of formic acid (122 μL, 32 equiv.) was added, and the reaction mixture was stirred for 2 h in open air. The resulting mixture was diluted with water (4 mL) and extracted with ethyl acetate (3 mL × 3). The solvent was removed under reduced pressure and the crude residue underwent ^1^H NMR yield determination. For the reactions with compete conversions, concentration of the organic phase under reduced pressure after drying over anhydrous sodium sulfate delivered desired products in quantitative yields.

Reaction Conditions E: α,β-unsaturated carbonyl compounds (0.1 mmol) and 0.4 mL of catalyst **C1** solution (0.01 mol/L) in deionized water were added to a 10 mL reaction tube. The mixture was stirred for 5 min at 80 °C, and then formic acid (122 μL, 32 equiv.) was added. After stirring for 1 h, a second portion of formic acid (122 μL, 32 equiv.) was added, and the reaction mixture was stirred for 2 h in open air. The resulting mixture was diluted with water (4 mL) and extracted with ethyl acetate (3 mL × 3). The solvent was removed under reduced pressure and the crude residue underwent ^1^H NMR yield determination.

#### 3.7.1. N-Propanoyl-2-Oxazolidinone (**2a**)

Performed on 0.14 mmol scale. Isolated by extraction, drying, and concentration. Yield: 21 mg, >99%, (*S/C* = 1400, 16 equiv. HCO_2_H, 1 h).

^1^H NMR (400 MHz, CDCl_3_) δ 4.42 (dd, *J* = 8.6, 7.6 Hz, 2H), 4.03 (t, *J* = 8.0 Hz, 2H), 2.94 (q, *J* = 7.4 Hz, 2H), 1.17 (t, *J* = 7.4 Hz, 3H); ^13^C NMR (101 MHz, CDCl_3_) δ 174.2, 153.6, 62.0, 42.5, 28.7, 8.2.

#### 3.7.2. Propionamide (**2b**)

Yield: 21% under Reaction Conditions B;

Performed on 0.1 mmol scale. Yield: 8 mg, >99% under Reaction Conditions D;

^1^H NMR (400 MHz, CDCl_3_) δ 5.72 (brs, 1H), 5.46 (brs, 1H), 2.27 (q, *J* = 7.6 Hz, 2H), 1.17 (t, *J* = 7.6 Hz, 3H); ^13^C NMR (101 MHz, CDCl_3_) δ 176.5, 28.9, 9.6.

#### 3.7.3. N,N-Diethylpropionamide (**2c**)

Yield: 60% under Reaction Conditions C;

Yield: 13 mg, >99% under Reaction Conditions D;

^1^H NMR (400 MHz, CDCl_3_) δ 3.38 (q, *J* = 7.2 Hz, 2H), 3.30 (q, *J* = 7.2 Hz, 2H), 2.32 (q, *J* = 7.4 Hz, 2H), 1.14 (m, 9H); ^13^C NMR (101 MHz, CDCl_3_) δ 172.8, 41.8, 40.0, 26.2, 14.3, 13.1, 9.6.

#### 3.7.4. N-Propionylpyrrolidine (**2d**)

Yield: 58% under Reaction Conditions C;

Yield: 13 mg, >99% under Reaction Conditions D;

^1^H NMR (400 MHz, CDCl_3_) δ 3.47 (t, *J* = 7.0 Hz, 2H), 3.40 (t, *J* = 6.8 Hz, 2H), 2.28 (q, *J* = 7.5 Hz, 2H), 2.01–1.90 (m, 2H), 1.89–1.79 (m, 2H), 1.15 (t, *J* = 7.5 Hz, 3H); ^13^C NMR (101 MHz, CDCl_3_) δ 172.3, 46.4, 45.6, 27.9, 26.1, 24.4, 9.0.

#### 3.7.5. N,N-Dibenzylpropionamide (**2e**)

Yield: 14% under Reaction Conditions C;

Yield: 47% under Reaction Conditions D.

^1^H NMR (400 MHz, CDCl_3_) δ 7.40–7.12 (m, 10H), 4.61 (s, 2H), 4.44 (s, 2H), 2.44 (q, *J* = 7.4 Hz, 2H), 1.20 (t, *J* = 7.4 Hz, 2H).

#### 3.7.6. N-Methylpropionanilide (**2f**)

Yield: 18% under Reaction Conditions C;

Yield: 50% under Reaction Conditions D;

^1^H NMR (400 MHz, CDCl_3_) δ 7.45–7.38 (m, 2H), 7.37–7.31 (m, 1H), 7.22–7.15 (m, 2H), 3.27 (s, 3H), 2.08 (q, *J* = 7.5 Hz, 2H), 1.05 (t, *J* = 7.5 Hz, 3H).

#### 3.7.7. N,N-Diphenylpropionamide (**2g**)

Yield: 53% under Reaction Conditions C;

Yield: 24 mg, >99% under Reaction Conditions D;

^1^H NMR (400 MHz, CDCl_3_) δ 7.49–7.08 (m, 10H), 2.27 (q, *J* = 7.4 Hz, 2H), 1.13 (t, *J* = 7.4 Hz, 3H); ^13^C NMR (101 MHz, CDCl_3_) δ 173.9, 142.9, 129.2, 126.5 (m), 28.7, 9.6.

#### 3.7.8. Isobutyramide (**2h**)

Yield: 10% under Reaction Conditions A;

Yield: 9 mg, >99% under Reaction Conditions D;

^1^H NMR (400 MHz, CDCl_3_) δ 5.52 (brs, 2H), 2.43 (hept, *J* = 6.9 Hz, 1H), 1.19 (d, *J* = 6.9 Hz, 6H); ^13^C NMR (101 MHz, CDCl_3_) δ 179.5, 35.0, 19.6.

#### 3.7.9. 3-Phenylpropanamide (**2i**)

Yield: 3% under Reaction Conditions A;

Yield: 42% under Reaction Conditions D.

^1^H NMR (400 MHz, CDCl_3_) δ 7.32–7.26 (m, 2H), 5.84 (brs, 1H), 5.48 (brs, 1H), 7.24–7.18 (m, 3H), 2.97 (t, *J* = 7.8 Hz, 2H), 2.53 (dd, *J* = 8.5, 7.0 Hz, 2H).

#### 3.7.10. Pyrrolidine-2,5-Dione (**2j**)

Yield: 50% under Reaction Conditions B;

Yield: 10 mg, >99% under Reaction Conditions D;

^1^H NMR (400 MHz, CDCl_3_) δ 8.41 (brs, 1H), 2.77 (s, 4H); ^13^C NMR (101 MHz, CDCl_3_) δ 177.4, 29.6.

#### 3.7.11. 3-Phenylpropanoic Acid (**2k**)

Yield: 7% under Reaction Conditions C;

Yield: 81% under Reaction Conditions E.

^1^H NMR (400 MHz, CDCl_3_) δ 10.34 (s, 1H), 7.34–7.27 (m, 2H), 7.25–7.18 (m, 3H), 2.97 (t, *J* = 7.8 Hz, 2H), 2.69 (t, *J* = 7.8 Hz, 2H).

#### 3.7.12. 2-Phenylpropionic Acid (**2l**)

Yield: 16 mg, >99% under Reaction Conditions C;

^1^H NMR (400 MHz, CDCl_3_) δ 11.17 (bs, 1H), 7.53–7.18 (m, 5H), 3.74 (q, *J* = 7.1 Hz, 1H), 1.51 (d, *J* = 7.1 Hz, 3H); ^13^C NMR (101 MHz, CDCl_3_) δ 180.5, 139.7, 128.7, 127.6, 127.4, 45.3, 18.1.

#### 3.7.13. Ethyl 3-Phenylpropanoate (**2m**)

Yield: 5% under Reaction Conditions C;

Yield: 47% under Reaction Conditions E.

^1^H NMR (400 MHz, CDCl_3_) δ 7.32–7.26 (m, 2H), 7.24–7.15 (m, 3H), 4.13 (q, *J* = 7.2 Hz, 2H), 2.95 (t, *J* = 7.9 Hz, 2H), 2.62 (dd, *J* = 8.5, 7.2 Hz, 2H), 1.23 (t, *J* = 7.1 Hz, 3H).

#### 3.7.14. Diethyl 2-(2,4-Dimethoxybenzyl)malonate (**2n**)

Yield: 30 mg, 96% under Reaction Conditions D.

^1^H NMR (400 MHz, CDCl_3_) δ 7.02 (d, *J* = 8.2 Hz, 1H), 6.42 (d, *J* = 2.4 Hz, 1H), 6.37 (dd, *J* = 8.3, 2.4 Hz, 1H), 4.13 (qq, *J* = 7.2, 3.6 Hz, 4H), 3.80 (s, 3H), 3.77 (s, 3H), 3.78 (t, *J* = 8.0 Hz, 1H), 3.13 (d, *J* = 8.0 Hz, 2H), 1.20 (t, *J* = 7.1 Hz, 6H).

#### 3.7.15. Dimethyl Succinate (**2o**)

Yield: 18% under Reaction Conditions C;

Yield: 68% under Reaction Conditions E. ^1^H NMR (400 MHz, CDCl_3_) δ 3.70 (s, 6H), 2.64 (s, 4H).

#### 3.7.16. 3-Phenylpropanenitrile (**2p**)

Yield: 2% in Reaction Conditions C;

Yield: 11% under Reaction Conditions E.

^1^H NMR (400 MHz, CDCl_3_) δ 7.38–7.34 (m, 2H), 7.33–7.30 (m, 1H), 7.27–7.24 (m, 2H), 2.99 (t, *J* = 7.4 Hz, 2H), 2.65 (t, *J* = 7.4 Hz, 2H).

### 3.8. Applications in Structural Modifications of Medicinal Molecules [35]

Santonin (50 mg, 0.2 mmol), MeCN (0.2 mL), and catalyst **C1** solution (0.02 mol/L, 0.2 mL) in deionized water were added to a 10 mL reaction tube. The mixture was stirred for 5 min at 80 °C, and then HCO_2_H (244 μL, 32 equiv.) was added. After stirring for 4 h in open air, saturated solution of NaHCO_3_ (15 mL) was added. The mixture was extracted with EA (10 mL × 3), dried over NaSO_4_, and concentrated in vacuo. The crude product was purified by flash column chromatography (PE/EA = 5:1, *v/v*) to give 1,2-dihydrosantonin (23 mg, 46%) as a colorless liquid. R*_f_* = 0.4 (PE/EA = 2:1, *v/v*); ^1^H NMR (400 MHz, CDCl_3_) δ 4.69 (d, *J* = 11.2 Hz, 1H), 2.60–2.30 (m, 3H), 2.01 (s, 3H), 1.99–1.50 (m, 7H), 1.34 (s, 3H), 1.27 (d, *J* = 6.8 Hz, 6H). ^13^C NMR (101 MHz, CDCl_3_) δ 198.8, 177.7, 152.5, 128.6, 81.9, 52.9, 41.7, 41.2, 38.3, 38.2, 33.6, 24.6, 24.2, 12.4, 11.2. The data are identical to those reported [51].

Compounds **4b–d** were synthesized in our previous work [35]. The data for **4b–d** are reported in our previous work [35].

### 3.9. Deuterium Substitution Experiments

3-acryloyloxazolidin-2-one (20 mg, 0.14 mmol) and 0.4 mL of catalyst solution (0.00025 mol/L, **C1**) in H_2_O or D_2_O were added to a 10 mL reaction tube. The mixture was stirred for 5 min at 80 °C, and then HCO_2_H, HCO_2_D, or DCO_2_D (3.0 equiv.) was added. After stirring for 1 h in open air, the mixture was diluted with water (4 mL) and extracted with ethyl acetate (3 mL × 3), followed by the of internal standard 1,3,5-trimethoxybenzene (7.9 mg, 0.047 mmol, 0.33 equiv). The organic solvent was evaporated under reduced pressure and the crude residue underwent ^1^H NMR to determine the yield and deterium incorporation.

### 3.10. Gram-Scale Experiment

3-acryloyloxazolidin-2-one (860 mg, 6.1 mmol), **C1** (2.5 mg, *S/C* = 1400), and 17 mL deionized water were sequentially added to a 25 mL round-bottom flask. The mixture was stirred for 5 min at 80 °C in an oil bath, and then formic acid (5.52 mL, 24 equiv.) was added. The reaction process was monitored by TLC. It took 2 h to reach complete consumption of 3-acryloyloxazolidin-2-one. The mixture was diluted with saturated Na_2_CO_3_ solution until no gas evolved. Extraction with ethyl acetate (10 mL × 3), drying over anhydrous Na_2_SO_4_, and rotatory evaporation under reduced pressure afforded the desired product as a white solid (790 mg, 91%). GC-MS indicates a good purity.

## 4. Conclusions

In summary, we have provided an iridium-catalyzed highly selective 1,4-reduction of α,β-unsaturated carbonyl compounds. A new iridium catalyst featuring a 2-(4,5-dihydroimidazol-2-yl)quinoline ligand shows high activity. Water can be used as solvent, and formic acid as the hydride source. The chemoselectivity is closely associated with the substrate structures. For α,β-unsaturated amides, acids, and esters, only the electron-deficient C=C bonds are reduced, and the other functional groups are left intact. However, for α,β-unsaturated ketones and aldehydes, 1,4-reduction occurs more feasibly than 1,2-reduction, but over reduction of the resulting ketone product is also observed. The catalyst efficiency is highly dependent on the electronic and steric properties of the substrates. For 3-acryloyloxazolidin-2-one, the *S/C* ratio is as high as 7000, and the initial TOF reaches a climax of 18,480 h^−1^. Deuterium labeling experiments show that the hydrogen of formyl group of formic acid finally transfers to the β-positions of the products. We also have performed a gram-scale 1,4-reduction, and demonstrated the application of the 1,4-reduction in the modification of pharmaceutical structures.

## Data Availability

Data are contained within the article.

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
