# Peer review of "Iridium-Catalyzed Highly Selective 1,4-Reduction of α,β-Unsaturated Carbonyl Compounds"

_molecules, 2024, doi:10.3390/molecules29245912_

Round 1

Reviewer 1 Report

Comments and Suggestions for Authors

In this paper, Yang and co-workers describe an iridium-catalyzed selective 1,4-reduction of α,β-unsaturated carbonyl compounds with formic acid as hydride donor in water. The protocol affords an efficient route to access diverse ketones, amides, esters and acids with general good yields and TOF values. From the viewpoint of green synthesis, this transformation is of interest and significance. After comprehensive consideration, the reviewer would like to recommend it for publication in Molecules with some minor revisions as below:

1.    Generally, the neutral conditions should be more attractive in the catalytic synthesis due to much better tolerance of functional groups. Why the authors emphasize the acidic condition in the introduction and which advantages are shown in the current method compared with previous reports?

2.     In the case of substrates containing internal alkenes, it seems that low efficience is observed, such as 2i, 2m and 1p, and some comments, explanations or potential side products should be reflected in the text.

3.    Given that the method is developed using water as solvent, D-containing formic acid and water should be employed to preliminarily investigate the mechanism.

4.    In scheme 4, why does the catalyst loading and the equivalent of formic acid need to changed constantly to perform this gram-scale synthesis?

5.    When a large amount of HCOOH is used in tube (seal?) at heating conditions, potential danger should be highlighted in experimental procedure.

6.    Please carefully examine all of the information, some non-scientific errors exist. For example, in lines 34 and 35, “a highly efficient TH 1,4-reduction in water under acidic conditions in water is still in demand”, a “in water” should be removed.

Author Response

Recommendation: Accept after minor review

Comments:

In this paper, Yang and co-workers describe an iridium-catalyzed selective 1,4-reduction of α,β-unsaturated carbonyl compounds with formic acid as hydride donor in water. The protocol affords an efficient route to access diverse ketones, amides, esters and acids with general good yields and TOF values. From the viewpoint of green synthesis, this transformation is of interest and significance. After comprehensive consideration, the reviewer would like to recommend it for publication in Molecules with some minor revisions as below:

  1. Our Generally, the neutral conditions should be more attractive in the catalytic synthesis due to much better tolerance of functional groups. Why the authors emphasize the acidic condition in the introduction and which advantages are shown in the current method compared with previous reports?

Our response:  At the end of the first paragraph, we add: “, because of the following facts: (1) The metal hydride reduction (for example NaBH4, LiAlH4) are often conducted under basic conditions, being incompatible with base-sensitive functionality such as carboxylic acid; (2) Compared with the neutral conditions, the acidic conditions can further activate the substrate and thus facilitate the reductions.”

  1. In the case of substrates containing internal alkenes, it seems that low efficience is observed, such as 2i, 2m and 1p, and some comments, explanations or potential side products should be reflected in the text.

Our response:  We add to the revised manuscript: “As shown by the 1H NMR of the crude reaction mixtures, in all the reactions with incomplete conversions (2i, 2m, 1p), no side-products were observed, and only unreacted substrate and product were observed.”

  1. Given that the method is developed using water as solvent, D-containing formic acid and water should be employed to preliminarily investigate the mechanism.

Our response:  Deuterium experiments and proposed mechanism are added to the revised manuscript. See Scheme 5, Scheme 6, and related discussions.

  1. In scheme 4, why does the catalyst loading and the equivalent of formic acid need to changed constantly to perform this gram-scale synthesis?

Our response: A new gram-scale reaction is performed to replace the previous one. We add: “The iridium-catalyzed 1,4-reduction is easy to scale up. For example, 0.86 g of 1a was readily reduced at 1400 S/C ratio in an open flask, and the product 2a was isolated in 91% yield by extraction with ethyl acetate, drying, and concentration under reduced pressure. GC-MS analysis shows good purity of the product.”

  1. When a large amount of HCOOH is used in tube (seal?) at heating conditions, potential danger should be highlighted in experimental procedure.

Our response: All the reactions were performed in a reaction tube or flask in open air. We have already mentioned this in the experimental procedures and in Scheme 4.

  1. Please carefully examine all of the information, some nonscientific errors exist. For example, in lines 34 and 35, “a highly efficient TH 1,4-reduction in water under acidic conditions in water is still in demand”, a “in water” should be removed.

Our response: Removed as suggested.

Reviewer 2 Report

Comments and Suggestions for Authors

In the manuscript „Iridium-catalyzed highly selective 1,4-reduction of alpha,beta-unsaturated carbonyl compounds with formic acid in water” the authors describe the application of an earlier reported iridium quinoline complex for the hydrogenation of various electron-deficient alkenes. The catalyst system is very effective and low loadings are possible when formic acid is used as reductant.

The conjugate reduction of polarized electron-deficient alkenes is an important research area. Although both the application of iridium complexes as well as formic acid are well established, the high efficiency and the selectivity of the reported Ir catalyst are interesting. However, for a new preparative method the reader may be interested more in the isolated yields of the products rather than the NMR-conversions. Once these data are provided the manuscript may become suitable for publication. In its current state publication seems premature.

Author Response

Referee: 2

Recommendation: Published after some clarifications

Comments:

In the manuscript „Iridium-catalyzed highly selective 1,4-reduction of alpha,beta-unsaturated carbonyl compounds with formic acid in water” the authors describe the application of an earlier reported iridium quinoline complex for the hydrogenation of various electron-deficient alkenes. The catalyst system is very effective and low loadings are possible when formic acid is used as reductant.

The conjugate reduction of polarized electrondeficient alkenes is an important research area. Although both the application of iridium complexes as well as formic acid are well established, the high efficiency and the selectivity of the reported Ir catalyst are interesting.

However, for a new preparative method the reader may be interested more in the isolated yields of the products rather than the NMR conversions. Once these data are provided the manuscript may become suitable for publication. In its current state publication seems premature.

Our response:  The manuscript has been substantially revised. For your concern, in the experimental section, we add: “For the reactions with compete conversions, concentration of the organic phase under reduced pressure after drying over anhydrous sodium sulfate delivered desired products in quantitative yields.” The isolated mass and percent yields for substrate scope and gram reactions are given.

Reviewer 3 Report

Comments and Suggestions for Authors

The manuscript describes a novel Ir catalyst for the selective transfer hydrogenation of double bonds in alpha-beta unsaturated carbonyl compounds. Indeed, this is still an area where improvement is necessary. The catalyst the authors present in this paper seems to be advantageous although there are still examples of unwanted 1,2-reduction of the carbonyl group.

I recommend the following corrections and additions:

1) Table 1 should be presented entirely on one page

2) The purity of all new compounds should be at least proven by HRMS

3) The carbon NMR signals of all compounds should be assigned.

4) On page 13, the respective numbers (see scheme 3) should be added

5) Why is only 4a described in the experimental part but not 4b-d? Please add the missing compounds.

6) The specific rotation should be added for all chiral compounds.

7) why is chapter 5 named "Patents" when it deals with supplementary material?

8) where is the supplemental material?

In general I recommend publication of the manuscript when my recommendations are considered.

Author Response

Referee: 3

Recommendation: Accept

Comments:

The manuscript describes a novel Ir catalyst for the selective transfer hydrogenation of double bonds in alphabeta unsaturated carbonyl compounds. Indeed, this is still an area where improvement is necessary. The catalyst the authors present in this paper seems to be advantageous although there are still examples of unwanted 1,2-reduction of the carbonyl group. I recommend the following corrections and additions:

  1. Table 1 should be presented entirely on one page.

Our response: Revised as suggested.

  1. The purity of all new compounds should be at least proven by HRMS.

Our response:  All the products are known compounds.

  1. The carbon NMR signals of all compounds should be assigned.

Our response: We think that the assignment of the carbon signals depends on empirical rule, and sometimes may be wrong. In order to not mislead readers (we cannot guarantee that we are 100% right), we decide not to give the assignment, as the ACS journal The Journal of Organic Chemistry requires. However, in the Materials and Instruments section, we add: “All the materials and products are known compounds, and their 1H and/or 13C NMR are identical with those reported.”

  1. On page 13, the respective numbers (see scheme 3) should be added.

Our response: Thank you for your comments and I have revised it in the paper.

  1. Why is only 4a described in the experimental part but not 4b-d? Please add the missing compounds.

Our response:  We add: “Compounds 4b-d are synthesised in ourpreviosu work [35]. The data of 4b-d are reported in our previous work.[35]

  1. The specific rotation should be added for all chiral compounds.

Our response:  We add: “The data are identical with those reported [51].” ”The data of 4b-d are reported in our previous work.[35]

  1. why is chapter 5 named "Patents" when it deals with supplementary material?

Our response: Thank you for your comments and I have deleted it.

  1. where is the supplemental material?

Our response:  We've uploaded the supporting material.

Reviewer 4 Report

Comments and Suggestions for Authors

The manuscript of Yang and co-workers deals about the selective 1,4-reduction of α,β-unsaturated carbonyl compounds using iridium catalysts. The authors presented a wide number of substrates, but the discussion of the results in function of the electronic end steric properties often results incomplete. The manuscript is not well-written and lack of concern in typesetting is evident. Before publication, I recommend to proofread the entire manuscript.

In particular:

a) The literature should be updated: for example, introducing in the appropriate context: J. Am. Chem. Soc. 2018, 140, 2, 606–609; Org. Lett. 2024, 26, 20, 4173–4177; Synlett 2022, 34(12): DOI: 10.1055/s-0042-1752986; RSC Adv., 2020, 10, 33706-33717.

b) It’s not clear where the paragraph 2. Results and Discussion starts. The introduction is too short and lack of information regarding the state of the art (see the request to insert additional references with explanation). 

c) The term “Recently” (line 36) for the auto citations is not appropriate (ref 10).

d) One typo: “in water” line 35 is repeated.

e) The following sentences are not clear and should be re-written: 

-line 88: “Excitingly,…..conditions”

-line 94 “Inert…..1q”

-line105-111: “the formation…..previous report” : the meaning of the entire sentences is not clear.

f) In “Materials and Methods”:

-The general procedure for scale up should be added.

-the characterization information of compounds 3b/3c and 3d should be added.

g) Paragraph 5. Patents should be deleted.

h) Supporting information files are not available. However, in “Supporting information” paragraph (in the manuscript, line 385-387), 19F NMR and HPLC spectra of products doesn’t make sense because there isn’t fluorine in the compounds and the authors declared to obtain the yield values by NMR-spectra.

In addition, the NMR spectra for C1 and C4 should be added to “supporting information”.

i) Uniform ref 11 c.

Author Response

Referee: 4

Recommendation: Published after some clarifications

Comments:

The manuscript of Yang and co-workers deals about the selective 1,4-reduction of α,β-unsaturated carbonyl compounds using iridium catalysts. The authors presented a wide number of substrates, but the discussion of the results in function of the electronic end steric properties often results incomplete. The manuscript is not wellwritten and lack of concern in typesetting is evident. Before publication, I recommend to proofread the entire manuscript.

  1. The literature should be updated: for example, introducing in the appropriate context: J. Am. Chem. Soc. 2018, 140, 2, 606–609; Org. Lett. 2024, 26, 20, 4173–4177; Synlett 2022, 34(12): DOI: 10.1055/s-0042-1752986; RSC Adv., 2020, 10, 33706-33717.

Our response: They are cited as refs. 27, 22, 28, 17.

  1. It’s not clear where the paragraph 2. Results and Discussion starts. The introduction is too short and lack of information regarding the state of the art (see the request to insert additional references with explanation).

Our response: We add the title “2. Results and Discussion” to show where it starts. According to your constructive suggestion, the introduction is improved. The recent state of the art are highlighted.

  1. The term “Recently” (line 36) for the auto citations is not appropriate (ref 10).

Our response: New recent reports from our group and others were cited. See refs. 31-43.

  1. One typo: “in water” line 35 is repeated.

Our response: Corrected.

  1. The following sentences are not clear and should be re-written:

-line 88: “Excitingly,…..conditions”

-line 94 “Inert…..1q”

-line105-111: “the formation…..previous report” : the meaning of the entire sentences is not clear.

Our response: Re-written as suggested.

  1. In “Materials and Methods”:

-The general procedure for scale up should be added.

-the characterization information of compounds 3b/3c and 3d should be added.

Our response: The general procedure for scale up are added. In the experimental section, we add: “Compounds 4b-d are synthesised in ourpreviosu work [35]. The data of 4b-d are reported in our previous work.[35]

  1. Paragraph 5. Patents should be deleted.

Our response: Deleted.

  1. Supporting information files are not available. However, in “Supporting information” paragraph (in the manuscript, line 385-387), 19F NMR and HPLC spectra of products doesn’t make sense because there isn’t fluorine in the compounds and the authors declared to obtain the yield values by NMR-spectra. In addition, the NMR spectra for C1 and C4 should be added to “supporting information”.

Our response: Supplementary materials are provided, and the information are provided. In “3.1. Materials and Instruments”, we add: “The catalysts (C2-C8) were prepared according to our previous publications [32-33].”

  1. Uniform ref 11 c.

Our response: Reformatted.

Reviewer 5 Report

Comments and Suggestions for Authors

The article reports a green method for the selective reduction of α, β-unsaturated carbonyl compounds. The article needs some formatting changes and some existing problems need to be solved before publication. The proposed revisions are as follows:

1.        Page 2, in the second paragraph, line fourInitial screening of the catalysts (entries 1-11) at an S/C ratio … Decreas- 56 ing the catalyst loading of C1 delivered lower yields (entries 17 and 18). A space needs to be added after the period between the first and second sentences. Please check all the main text.

2.        Pages 3 and 4, it is suggested to expand the substrate scope to include alkyl chain alkenes, halogenated alkenes, heterocyclic alkenes, and other related compounds.

3.        Page 10, reaction condition D includes the addition of formic acid in two distinct steps, different from the other conditions. Whether this two-step addition of formic acid contributes to the enhanced yield.

4.        Page 10, reaction condition E was not previously illustrated in the text. If this condition affects the yield, add the yield under this condition on page 4; if there is no significant effect, remove it.

5.        It's suggested to add a new section in the paper that clearly explains the reaction mechanism.

Author Response

Referee: 5

Recommendation: Published after some clarifications

Comments:

The article reports a green method for the selective reduction of α, β-unsaturated carbonyl compounds. The article needs some formatting changes and some existing problems need to be solved before publication. The proposed revisions are as follows:

  1. Page 2, in the second paragraph, line four“Initial screening of the catalysts (entries 1-11) at an S/C ratio … Decreas- 56 ing the catalyst loading of C1 delivered lower yields (entries 17 and 18).” A space needs to be added after the period between the first and second sentences. Please check all the main text.

Our response: Added. Similar errors have been corrected through out the manuscript.

  1. Pages 3 and 4, it is suggested to expand the substrate scope to include alkyl chain alkenes, halogenated alkenes, heterocyclic alkenes, and other related compounds.

Our response: Thank you for your constructive suggestion. The current scope is not applicable to those substrates.

  1. Page 10, reaction condition D includes the addition of formic acid in two distinct steps, diferent from the other conditions. Whether this twostep addition of formic acid contributes to the enhanced yield.

Our response: Above Figure 1, we add: “For Conditions D and E, the formic acid was added portion-wise, because in our previous work we found that the catalyst efficiency will be lowered by strong acidity that was rendered by large amount of formic acid. By portion-wise addition, the second portion was added following consumption of the first portion, in this way keeping the reaction media less acidic.”

  1. Page 10, reaction condition E was not previously illustrated in the text. If this condition affects the yield, add the yield under this condition on page 4; if there is no significant effect, remove it.

Our response: Conditions E was only applied to 1p.

  1. It's suggested to add a new section in the paper that clearly explains the reaction mechanism.

Our response: We conducted deuterium labelling experiments to study the mechanism and the mechanism is added.

Reviewer 6 Report

Comments and Suggestions for Authors

This is clearly a "results" paper since all that is presented are the conditions used and yields of products for a variety of HT hydrogenations. Despite a comment early on that the effectiveness of the hydrogenation depends on the nature and structure of the substrate, there doesn't appear to be any discussion of this point nor is there any comment on the mechanism (has it been established?) and how that might tie in with this. I would like to see some discussion of this before I can recommend publication.

Author Response

Referee: 6

Recommendation: Published after some clarifications

Comments:

This is clearly a "results" paper since all that is presented are the conditions used and yields of products for a variety of HT hydrogenations. Despite a comment early on that the effectiveness of the hydrogenation depends on the nature and structure of the substrate, there doesn't appear to be any discussion of this point nor is there any comment on the mechanism (has it been established?) and how that might tie in with this. I would like to see some discussion of this before I can recommend publication.

Our response: The manuscript has been substantially revised, according to all the referees’ comments. We conducted deuterium labelling experiments to study the mechanism and the mechanism is added.

Round 2

Reviewer 2 Report

Comments and Suggestions for Authors

The authors have revised the manuscript and addressed the questions raised by the referees. Thereby, the quality of the paper has been improved. 

However, before publication can be recommended, the authors may want to include clean NMR spectra of the products isolated. In particular, the 1H NMR spectra of products 2b and 2d show impurities.

Author Response

Comments and Suggestions for Authors:

The authors have revised the manuscript and addressed the questions raised by the referees. Thereby, the quality of the paper has been improved. However, before publication can be recommended, the authors may want to include clean NMR spectra of the products isolated. In particular, the 1H NMR spectra of products 2b and 2d show impurities.

Our response: We have repeated the reactions and obtained pure products 2b and 2d. The yields of 2b and 2d remain the same. Their 1H and 13C NMR spectra are updated in the supplementary materials.

Reviewer 4 Report

Comments and Suggestions for Authors

The manuscript was improved in a correct way.

Author Response

Comments and Suggestions for Authors:

The manuscript was improved in a correct way.

Our response: No need to response.

Reviewer 6 Report

Comments and Suggestions for Authors

Although the added discussion is minimal, it is enough to satisfy my original criticism.

Author Response

Although the added discussion is minimal, it is enough to satisfy my original criticism.

Our response: No need to response.